# Synergistic Anticancer Activity of HSP70 Inhibitor and Doxorubicin in Gain-of-Function Mutated p53 Breast Cancer Cells

**DOI:** 10.3390/biomedicines13051034

**Published:** 2025-04-24

**Authors:** Kuan-Yo Wu, Ana Crucho, Mia Su, Sih-Tong Chen, Chen-Hsiu Hung, Yu-Ling Kou, Yu-Jie Liu, Tzu-Chi Hsu, Fang-Yu Yeh, Ching-Feng Lien, Chia-Chi Chen, Bi-He Cai

**Affiliations:** 1Department of Biomedical Engineering, I-Shou University, Kaohsiung City 82445, Taiwan; a0932359630@gmail.com; 2Lisbon School of Medicine, Lisbon University, 1649-004 Lisboa, Portugal; anacrucho@edu.ulisboa.pt; 3Faculty of Medicine and Health Sciences, University of Sherbrooke, Sherbrooke, QC J1H 5N4, Canada; mia.su@usherbrooke.qc.ca; 4Department of Medical Science and Biotechnology, I-Shou University, Kaohsiung City 82445, Taiwan; 0613armychen@gmail.com; 5Department of Medical Laboratory Science, I-Shou University, Kaohsiung City 82445, Taiwan; joy0983321933@gmail.com (C.-H.H.); kullin1219@gmail.com (Y.-L.K.); kelly0909381362@gmail.com (Y.-J.L.); cinzia9405@gmail.com (T.-C.H.); 6School of Medicine, I-Shou University, Kaohsiung City 82445, Taiwan; lily920127@gmail.com (F.-Y.Y.); lien980206@yahoo.com.tw (C.-F.L.); 7Department of Otolaryngology-Head and Neck Surgery, E-Da Hospital, Kaohsiung City 82445, Taiwan; 8Department of Pathology, E-Da Hospital, Kaohsiung City 82445, Taiwan; 9Department of Physical Therapy, I-Shou University, Kaohsiung City 82445, Taiwan; 10Department of Occupational Therapy, I-Shou University, Kaohsiung City 82445, Taiwan

**Keywords:** breast cancer, p53 mutation, aggregation, HSP70 inhibitor, apoptosis

## Abstract

**Background**: The mutation rate of p53 in breast cancer is around 20%. Specific p53 mutations exhibit prion-like abnormal misfolding and aggregation and gain oncogenic function, causing resistance to the chemotherapy drug doxorubicin. In this study, we identified key upstream regulatory molecules that inhibit the aggregation of p53 with the aim of increasing the anticancer effect of doxorubicin. **Methods**: Thioflavin T was employed as a fluorescent probe to detect prion-like protein aggregates within cells, the response to various inhibitors was evaluated using CCK8 assay, and the coefficient of drug interaction was calculated. The cell apoptosis ratio was evaluated using Caspase-3/7 based flow cytometry assay. **Results**: MDA-MB-231 cells (with p53 R280K mutation) and T47D cells (with p53 L194F mutation) had a strong Thioflavin T staining signal, but MDA-MB-468 cells (with p53 R273H mutation) had a weak Thioflavin T signal. Compared to MDA-MB-468 cells, which had a good response to doxorubicin, both MDA-MB-231 and T47D showed high doxorubicin drug resistance. Co-treatment with various misfolding p53 aggregation inhibitors and doxorubicin found that only the HSP70 inhibitor and doxorubicin had synergistic anticancer activity in both MDA-MB-231 and T47D cells. Furthermore, this co-treatment induced cell apoptosis in MDA-MB-231, which was reversed by a pan-caspase inhibitor. **Conclusions**: Doxorubicin resistance caused by specific p53 mutants can be resolved by co-treatment with a HSP70 inhibitor in breast cancer cells.

## 1. Introduction

We learned from the TP53 Database (https://tp53.isb-cgc.org/ [accessed on 1 July 2024]) [1] that the p53 mutation rate in breast cancer patients is 22.8% (Figure 1A). The p53 mutation rate in breast cancer patients in Taiwan is exactly 20% (Figure 1B). Among samples with p53 point mutations, 84.92% were missense mutations, and 82.59% of missense mutations were non-functional p53 (Figure 1C). Therefore, p53 tumor suppressor gene mutations play a significant role in breast cancer. These mutations often lead to the loss of their tumor suppressor function (LOF), with the two downstream genes of p53, p21, and 14-3-3σ [2,3,4], which are brake genes that regulate the cell cycle at G1-S and G2-M, respectively, being particularly affected. Because p53 is unable to reactivate p21 and 14-3-3σ, the cell cycle becomes uncontrolled and the cells proliferate rapidly [2,3].

In addition to LOF-type p53 mutations, certain specific p53 mutations can lead to p53 aggregation. Aggregated mutant p53 can lead to p53 gain of function (GOF) [5,6]. GOF p53 can lead to metastasis, increased tumor growth, and the development of chemoresistance [7,8]. Thioflavin T (ThT) is used to stain the abnormal misfolding and aggregation of prion-like proteins in cells [5,9,10]. We previously found that several mutants of p53 (R175H and R280T) co-localize with ThT-stained aggregates in head and neck cancer [11,12], and specific mutants of p53 have also been found to co-localize with protein aggregates in breast cancer [13].

Doxorubicin (DOX) is a chemotherapeutic drug that can induce apoptosis in the breast cancer cell line MCF7 with p53 wild-type cells at a concentration of 1 μM for 24 h. After 24 h of treatment with 1 μM DOX, breast cancer cell lines T47D (with p53 L194F mutation) and CRL2324 (with p53 R175H mutation) showed no apoptotic response [10], indicating that breast cancer cells with specific p53 mutations are resistant to chemotherapy drugs. Therefore, in this study, we attempted to find the key upstream regulatory molecules that inhibit the aggregation of p53, which can be used to increase the anticancer effect of the chemotherapy drug doxorubicin.

## 2. Materials and Methods

### 2.1. Cell Culture and Drug Treatment

The breast cancer cell lines MDA-MB-231, MDA-MB-468, T47D, and MCF7 were from the Bioresource Collection and Research Center (BCRC), and the MDA-MB-468 cell line was obtained from the American Type Culture Collection (ATCC). These cell lines were cultured at 37 °C in 5% CO_2_. MDA-MB-231, MDA-MB-468, and MCF7 cells were grown in DMEM (Invitrogen, Carlsbad, CA, USA), and T47D cells were maintained in RPMI 1640 (Invitrogen). All media were supplemented with 10% fetal bovine serum (FBS) (Invitrogen), along with 100 U/mL penicillin and 100 μg/mL streptomycin (both from Invitrogen). Cells were treated with DMSO alone, 1 μM Doxorubicin (Sigma-Aldrich, St. Louis, MO, USA), 5 μM IPI-504 (HSP90 inhibitor, MedChemExpress, Monmouth Junction, NJ, USA), 10 μM VER-155008 (HSP70 inhibitor, MedChemExpress), 100 μM BAPN (LOX inhibitor, MedChemExpress), 5 μM FK886 (NAMPT inhibitor, MedChemExpress), or 20 μM z-VAD-FMK (pan-caspase inhibitor, MedChemExpress).

### 2.2. CCK8 Assay

Cell viability was assessed using the CCK-8 assay (Invitrogen). Cells treated with vehicle control (DMSO) served as the reference group and were defined as 100% viability for normalization purposes. For the assay, 10 μL of CCK-8 reagent was added to each well of a 96-well plate, followed by incubation at 37 °C in a humidified CO_2_ incubator for 1 h. Absorbance at 450 nm was then measured using a SpectraMax iD3 microplate reader (Molecular Devices, Silicon Valley, CA, USA). Wells containing medium without cells were used as blank controls. Cell viability (%) was calculated using the following formula: [(OD_(Drug)_ − OD_(Blank)_)/(OD_(Mock)_ − OD_(Blank)_)] × 100.

### 2.3. Drug Interaction Analysis

The drug interaction coefficient (CDI) was determined using the following formula: CDI = AB/(A × B) [14], where AB represents the ratio for the combination drug group based on cell viability from the CCK8 assay, and A and B refer to the ratios for the single drug groups. CDI values of <1, =1, or >1 represent synergistic, additive, or antagonistic drug interactions, respectively. Specifically, a CDI value of less than 0.7 indicates a significant synergistic effect [15].

### 2.4. Thioflavin T Staining

A 10 mM stock solution of Thioflavin T (Sigma-Aldrich) in water was diluted with 1X dPBS to achieve a final working concentration of 10 μM (1:1000). Similarly, a 1 mg/mL stock solution of Hoechst 33342 (AAT Bioquest, Sunnyvale, CA, USA) was diluted with 1X dPBS to obtain a final working concentration of 1 μg/mL (1:1000). Cancer cells (3000 per well) were seeded in a 96-well plate. After cell incubation for one day, the culture medium was removed, and 100 μL 1X dPBS and diluted Thioflavin T and Hoechst 33342 were added. After co-staining with both Thioflavin T and Hoechst 33342 for 30 min, the cells were washed three times with 1X dPBS. ThT fluorescence was measured using a SpectraMax iD3 microplate reader (Molecular Devices) with an excitation wavelength of 450 nm and an emission wavelength of 490 nm. Hoechst 33342 fluorescence was measured at an excitation wavelength of 360 nm and an emission wavelength of 460 nm. The protein aggregation signal was calculated as the ratio of ThT optical density to Hoechst 33342 optical density. Normalized ThT fluorescence intensity was set as the ratio of ThT optical density to Hoechst 33342 optical density in the DMSO-treated group, designated as 1.

### 2.5. Real-Time RT-PCR

Total cellular RNA was isolated and purified using TRIzol reagent (Invitrogen). To prepare cDNA from mRNAs, 1 μg of total RNA was reverse transcribed using the QuantiNova Reverse Transcription Kit (Qiagen, Hilden, Germany). The resulting cDNA samples were then combined with each of pairing primers and ready-to-use 5X-concentrated HOT FIREPol EvaGreen qPCR Mix Plus solution (Omics Bio, New Taipei City, Taiwan). The amplification of qPCR signal was monitored with the Quant Studio 3 System (Applied Biosystems, Waltham, MA, USA). Relative quantification of gene transcription levels was determined using the 2^−ΔΔCT^ method. The sequences of the forward (F) and reverse (R) pairing primers were as follows: GAPDH, F: GTCTCCTCTGACTTCAACAGCG and R: ACCACCCTGTTGCTGTAGCCAA; BAG2, F: AACGCTAAAGCCAACGAGGG and R: CAGCAGTTGCTGCTTCTCTCA; LOXL1, F: TGGTCCCAGACCCCAACTAT and R: GTAGCACCCGCACATCGTAG.

### 2.6. Flow Cytometry

Cultured cells (5 × 10^5^ per tube) were collected and centrifuged at 300× *g* for 5 min. Pelleted cells were then incubated with 1 µL of CellEvent Caspase-3/7 Green Detection Reagent (Invitrogen) (final concentration: 500 μM) and 1 µL of SYTOX AADvanced Dead Cell Stain (Invitrogen) (final concentration: 1 mM) for 30 min at 4 °C. Following incubation, cells were washed twice with 1 mL of FACS buffer (STEMCELL Technologies, Vancouver, BC, Canada) (PBS supplemented with 2% fetal calf serum), resuspended in 400 µL of FACS buffer, and maintained on ice in the dark until analysis. Prior to flow cytometry, samples were filtered through a cell strainer to remove clumps. Data acquisition was performed using a NovoCyte flow cytometer (Agilent, San Diego, CA, USA) and analysis was conducted with NovoExpress software (version 1.6.2) (Agilent, San Diego, CA, USA). The proportion of apoptotic cells was quantified based on double-staining profiles, identifying as caspase-3/7-positive and SYTOX-negative population.

### 2.7. Immunocytochemistry

The culture medium was aspirated from each well, followed by two washes with 1× PBS. A 100 μL volume of fixative solution (4% paraformaldehyde in 1× PBS) was then added to each well and incubated for 20 min. After fixation, wells were washed twice with 100 μL of wash buffer (1% BSA in 1× PBS). A 100 μL Blocking buffer (0.2% Triton X-100 (Sigma-Aldrich) and 1% BSA in 1× PBS) was added to each well and incubated for 30 min. The blocking buffer was then removed, and wells were washed three times with 100 μL of wash buffer. Subsequently, a staining mixture was prepared by diluting PE-conjugated mouse IgG2a secondary antibody (1:200), Hoechst 33342 (1 mg/mL stock; 1:1000), and thioflavin T (10 mM stock; 1:1000) in blocking buffer. A total of 100 μL of this mixture was added to each well, and the cells were incubated for 30 min in the dark. Finally, wells were washed three times with 100 μL of wash buffer. Fluorescence images were captured using an ECLIPSE Ts2 microscope (Nikon, Tokyo, Japan), and merged images were processed using ImageJ software (version 1.54) [16].

### 2.8. Activated Caspase-3/7 Detection Staining

CellEvent Caspase-3/7 Detection Reagents are composed of a DEVD peptide linked to a nucleic acid-binding dye. The DEVD sequence serves as a cleavage site for caspase-3/7, rendering the dye non-fluorescent until it is cleaved and bound to DNA. Once caspase-3/7 is activated in apoptotic cells, the peptide is cleaved, enabling the dye to bind to DNA and produce a strong fluorescence signal. To assess caspase-3/7 activity, CellEvent Caspase-3/7 Red Detection Reagent (Invitrogen) was prepared at a final concentration of 5 μM in PBS. Following removal of the culture medium, 100 μL of the diluted reagent was added to each well of a 96-well plate. Cells were incubated at 37 °C in a humidified CO_2_ incubator for 30 min to allow for caspase-mediated activation of the fluorogenic substrate. After incubation, nuclei were counterstained with Hoechst 33342. Fluorescent images were acquired using an ECLIPSE Ts2 fluorescence microscope (Nikon, Tokyo, Japan).

### 2.9. Statistical Analysis

Statistical analysis between two experimental groups was conducted using a two-tailed Student’s *t*-test. Results are expressed as mean values ± standard deviation (SD). Statistical significance was defined as a *p*-value less than 0.05, with significance levels indicated as follows: *, *p* < 0.05; **, *p* < 0.01; ***, *p* < 0.001.

## 3. Results

### 3.1. Doxorubicin Resistance Appears in Two High ThT-Staining Breast Cancer Cell Lines, MDA-MB-231 and T47D

First, we used ThT staining to observe intracellular protein aggregates in several breast cancer cell lines: MDA-MB-231 (containing p53 R280K mutation), MDA-MB-468 (containing p53 R273H mutation), T47D (containing p53 L194F mutation), and MCF7 (containing wild-type p53). The results showed that the signal strength of ThT was MDA-MB-231 > T47D > MDA-MB-468 ≈ MCF7 (Figure 2A,B). Therefore, two p53 mutant breast cancer cell lines, MDA-MB-231 and T47D, had more intracellular protein aggregates than the MDA-MB-468 with p53 R273H mutation. The co-localization of p53 and ThT was verified by co-staining in both T47D and MDA-MB-231 cells (Appendix A). We further used doxorubicin to treat all the p53-mutated breast cancer cell lines. After two days of treatment with 1 μM doxorubicin, the survival rate was MDA-MB-231 > T47D > MDA-MB-468 (Figure 3). Therefore, we concluded that the MDA-MB-231 and T47D with GOF-type p53 mutations are much more resistant to doxorubicin.

### 3.2. HSP70 Inhibitor and Doxorubicin Exhibit Synergistic Anticancer Activity in MDA-MB-231

We attempted to identify key upstream regulatory molecules that inhibit p53 aggregation as a possible means to enhance the anticancer effect of chemotherapy drugs. It has been found that BAG2 can bind to misfolded p53 mutants and promote the aggregation of misfolded mutant p53 through HSP90, thereby ensuring the increase and maintenance of aggregates [17]. BAG2-mediated mutant p53 aggregation inhibits the mitochondrial apoptosis pathway, leading to chemoresistance in breast cancer cells, and HSP90 inhibitor (IPI-504) can relieve BAG2-mediated mutant p53 aggregation [17]. LOXL1 was found to prevent the ubiquitination of BAG2 in glioma and stabilize BAG2 [18]. LOXL1 is a member of the lysyl oxidase family. The lysyl oxidase family has five members, namely LOX, LOXL1, LOXL2, LOXL3, and LOXL4 [19], among which only LOXL1 exhibited a markedly elevated expression in breast cancer tumor tissues relative to normal breast tissue [20]. In this study, we also found that LOXL1 and BAG2 were expressed in two mutant p53 cell lines, MDA-MB-231 and T47D, and expressions of these genes were higher than they were in wild-type p53 MCF7 cells (Figure 4).

In addition, our previous experiments found that HSP70 inhibitors and NAMPT inhibitors can effectively reduce the accumulation of misfolded mutant p53 in head and neck cancer [11]. We co-treated breast cancer cells MDA-MB-231 (with high LOXL1 and BAG2 expression) with HSP90 inhibitor (IPI-504), HSP70 inhibitor (VER-155008), LOX inhibitor (BAPN), or NAMPT (FK886) inhibitor and doxorubicin, and we observed the survival rate to calculate the drug interactions. The effect coefficient was calculated to find the optimal anticancer drug treatment combination to confirm which inhibitor of mutant p53 aggregation could effectively combat doxorubicin resistance. The results showed that co-treatment of doxorubicin and VER-155008 achieved the best cell cytotoxic effect (Figure 5A) and had the best drug synergistic cytotoxic effect (Figure 5B–E). We further assayed the ThT signaling under treatment with HSP90 inhibitor (IPI-504) or HSP70 inhibitor (VER-155008); both treatments reduced the signal strength of ThT in MDA-MB-231 cells (Figure 6).

### 3.3. HSP70 Inhibitor and Doxorubicin Have Synergistic Anticancer Activity in T47D, and HSP90 Inhibitor Has No Cytotoxic Reponse in T47D

HSP90 inhibitor (IPI-504) and HSP70 inhibitor (VER-155008) were also used in combination with doxorubicin to treat T47D cells. The synergistic anticancer activity of HSP70 inhibitor and doxorubicin was observed in T47D cells (Figure 7A,B). Surprisingly, T47D cells were much more resistant to IPI-504 than MDA-MB-231 (Figure 5A and Figure 7A).

### 3.4. Combination Therapy of HSP70 Inhibitor and Doxorubicin Induces Apoptosis in MDA-MB-231 Cells

The cytotoxic effect of anticancer drugs may lead to cell necrosis or apoptosis. In the execution phase of apoptosis, the activation of several caspases is a key mechanism [21,22,23]. Thus, we used the broad caspase inhibitor z-VAD-FMK to test whether the cell cytotoxicity induced by co-treatment with doxorubicin and HSP70 inhibitor (VER-155008) could be reversed. It was found that z-VAD-FMK did indeed reverse the cytotoxic effect of cells co-treated with doxorubicin and VER-155008, significantly increasing the cell survival rate (Figure 8A). Therefore, we concluded that co-treatment of breast cancer cells with doxorubicin and VER-155008 did induce apoptosis.

Finally, we used flow cytometry to analyze the effects of doxorubicin, VER-155008, or their combination on tumor cell apoptosis in MDA-MB-231 cells. We found that combination treatment of the two drugs increased the apoptosis rate of MDA-MB-231 tumor cells induced by doxorubicin or VER-155008 alone by 27% and 136%, respectively (Figure 8B). In addition, we found that co-treatment with doxorubicin and VER-155008 dramatically activated caspase3/7 in T47D cells (Appendix A).

The findings from the CCK-8 survival assay and the caspase inhibitor assay suggest that cell apoptosis is the primary mechanism by which doxorubicin and VER-155008 induce the death of breast cancer cells.

### 3.5. The Expression of hsp70 Family Members in Breast Invasive Carcinoma (BRCA)

Recently, two studies have reported that HSP70 inhibitors can reverse the doxorubicin resistance in breast cancer cells [24,25]. However, these studies did not investigate the aggregation of p53 GOF with the same combination drug treatments. Furthermore, we used gene expression databases to compare the expression profiles of different Hsp70 family members in wild-type and p53-mutated breast cancers. The Hsp70 family has three members in humans, HSPA1A, HSPA1B, and HSPA1L [26,27]. According to the GEPIA2 database [28], both HSPA1A and HSPA1B are significantly highly expressed in breast invasive carcinoma (BRCA) compared to healthy tissue (Figure 9A,B). However, HSPA1L exhibits relatively low expression in both normal and BRCA tissues (Figure 9C). According to the UALCAN database [29], HSPA1A and HSPA1B are highly expressed in both wild-type and p53-mutated BRCA compared to healthy tissue (Appendix A), but HSPA1A expression is relatively lower in the p53-mutated BRCA compared to BRCA with wild-type p53 (Appendix A).

## 4. Discussion

The HSP70 inhibitor VER-155008 decreases the ThT signal in the Detroit 562 cell line, a head and neck squamous cell carcinoma (HNSCC) cell line harboring the p53 R175H mutation [11]. SIRT1, an NAD+-dependent protein deacetylase, has been shown to deacetylate Heat Shock Factor 1 (HSF1), thereby promoting HSF1-mediated upregulation of HSP70 [30]. NAMPT can positively regulate SIRT1 activity by increasing NAD+ levels [31]. NAMPT inhibitor FK886 was also found to reduce the ThT signal in Detroit 562 cells [11]. Furthermore, the NAMPT inhibitor and p73 activator were able to repress Detroit 562 (containing p53 R175H) and HONE-1 (containing p53 R280T) HNSCC cell proliferation in a synergistic manner [11,12]. Here, we found that both HSP70 inhibitor (VER-155008) or NAMPT inhibitor (FK886) and doxorubicin were able to repress MDA-MB-231 (containing p53 R280K mutation) breast cancer cell proliferation in a synergistic manner (Figure 5C,E). Doxorubicin induces p63 protein expression in UMSCC10B (containing p53 G245C mutation) HNSCC cells [32]. Additionally, knocking down p63 partially reverses doxorubicin-induced apoptosis in UMSCC10B cells [32], indicating that p63 is critical for the apoptotic response to doxorubicin in HNSCC cells [32]. The RNA sequencing database of cell lines in the Human Protein Atlas showed that the average value of normalized transcripts per million (nTPM) of p63 in 38 different kinds of head and neck cancer cell lines was 125.5, but the average value nTPM of p63 in 62 different types of breast cancer cell lines was only 4.5 (Appendix A) [33]. The role of p63 in doxorubicin-mediated cell apoptosis in p53 mutated breast cancer cells, thus, needs further investigation. Aggregated forms of misfolded p53 mutants have been shown to exert dominant-negative effects by sequestering other tumor suppressor proteins, such as p63 and p73, thereby impairing their normal function [34,35]. To relieve the mutated p53 aggregation may make it much easier for p63 or p73 to liberate and release their anticancer role. Therefore, p63 or p73 activators may be valuable for use with misfolding p53 aggregation inhibitor(s) to achieve good anticancer effects.

HSP90 inhibitor (IPI-504) is known to repress the mutant p53 aggregates in both SK-BR-3 (containing p53 R175H) and BT-549 (containing p53 R249S) breast cancer cell lines, which have high BAG2 expression [17]. LOXL1 was found to prevent the ubiquitination of BAG2 and stabilize BAG2 [18]. In this study, MDA-MB-231 cells (containing p53 R280K mutation) had a good response to IPI-504 (Figure 5), but T47D cells (containing p53 L194F mutation) were highly resistant to IPI-504 (Figure 7). In addition, our data showed a much higher levels of expression of LOXL1 and BAG2 in MDA-MB-231 than T47D (Figure 4). The Human Protein Atlas also showed similar results [33]. In MDA-MB-231 cells, the nTPM value for BAG2 was 21.5 (Appendix A), but the value for T47D was only 1.9. Correspondingly, the nTPM value for LOXL1 was 5.5 within MDA-MB-231 cells, but only 0.5 for T47D (Appendix A). Therefore, breast cancer cells with relatively low expression of BAG2 and LOXL1 may have a high possibility of resistance to IPI-504.

Two other studies have reported the relief of doxorubicin resistance by HSP70 inhibitors in breast cancer, through the Mint3-HIF-1-HSP70 pathway and the DNAJC12-HSP70-AKT pathway [24,25]. In our previous study, we found that the NAMPT-SIRT1-HSP70 pathway could control the p73 activator anticancer activity in HNSCC cells containing GOF p53 mutants [11,36]. GOF-type p53 can also activate AKT to achieve the oncogenic phenomenon in HNSCC cells [37]. The mechanism through which the drug combination of the HSP70 inhibitor and doxorubicin achieve anticancer activity in breast cancer cells is very complex, and whether other pathways or molecules (Mint3 or DNAJC12) are related to p53 need to be investigated. Furthermore, the up- and down-stream molecule interaction should be investigated.

Although doxorubicin is effective at achieving anticancer effects in breast cancer patients [38,39], cardiotoxic effects of doxorubicin that induce cardiomyocyte death have also been reported [40,41]. A reduction in the dosage of doxorubicin for use in breast cancer therapy by combining it with other drugs may avoid such side effects. Furthermore, the synergistic cytotoxicity of breast cancer cells observed under HSP70 inhibitor and doxorubicin co-treatment detailed in this study may provide a feasible strategy for future preclinical studies, particularly for evaluating the toxic effects on normal cardiac cells.

## 5. Conclusions

HSP70 inhibitor and doxorubicin have synergistic anticancer activity in both MDA-MB-231 and T47D cells. Both of these cell lines harbor GOF-type p53 mutations that are highly resistant to doxorubicin. The specific p53 mutants used in this study showed that resistance to the chemotherapy drug doxorubicin will likely be resolved by administering it as a co-treatment together with HSP70 inhibitors in breast cancer for future potential application in clinical usage.

## Figures and Tables

**Figure 1 biomedicines-13-01034-f001:**
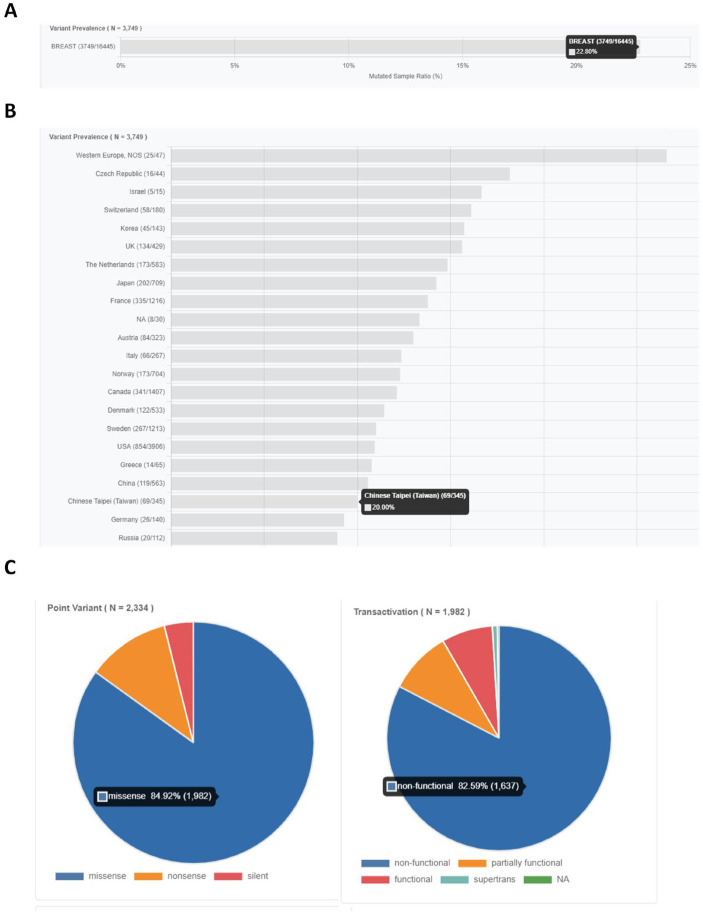
The frequency of p53 mutations in breast cancer patients according to the TP53 Database (https://tp53.isb-cgc.org/ [accessed on 1 July 2024]). The data show that (**A**) the mutation rate of p53 in breast cancer patients is 3749/16,445 (22.8%) and (**B**) the mutation rate of p53 in breast cancer patients in Taiwan is 20%. (**C**) Among samples harboring point mutations in the *TP53* gene, the majority (84.92%) were classified as missense mutations. Of these, 82.59% resulted in non-functional p53 protein, indicating a substantial loss of tumor suppressor activity in most cases.

**Figure 2 biomedicines-13-01034-f002:**
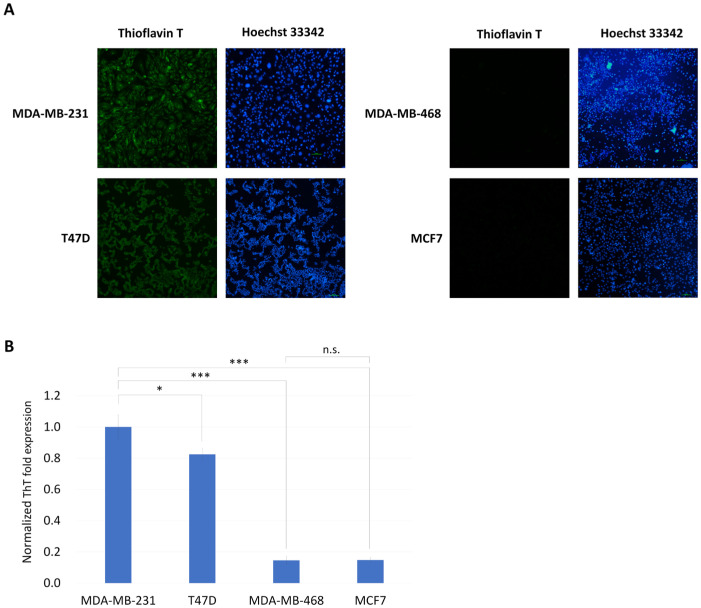
ThT signaling in various breast cancer cell lines. (**A**) ThT signals for MDA-MB-231, MDA-MB-468, T47D, and MCF7 cell lines. ThT was used to stain protein aggregates (shown in green), and Hoechst 33342 was used for nuclear counterstaining (shown in blue). Scale bar: 100 µm (20× magnification). (**B**) The OD ratio of ThT/Hoechst 33342, and with ThT intensity in MDA-MB-231, was set to 1 for normalization across the other cell lines. The results indicate that ThT intensity follows the order MDA-MB-231 > T47D > MDA-MB-468 ≈ MCF7. Data are presented as mean ± SD, n = 3 (*, *p* < 0.05; ***, *p* < 0.001; n.s., *p* > 0.05).

**Figure 3 biomedicines-13-01034-f003:**
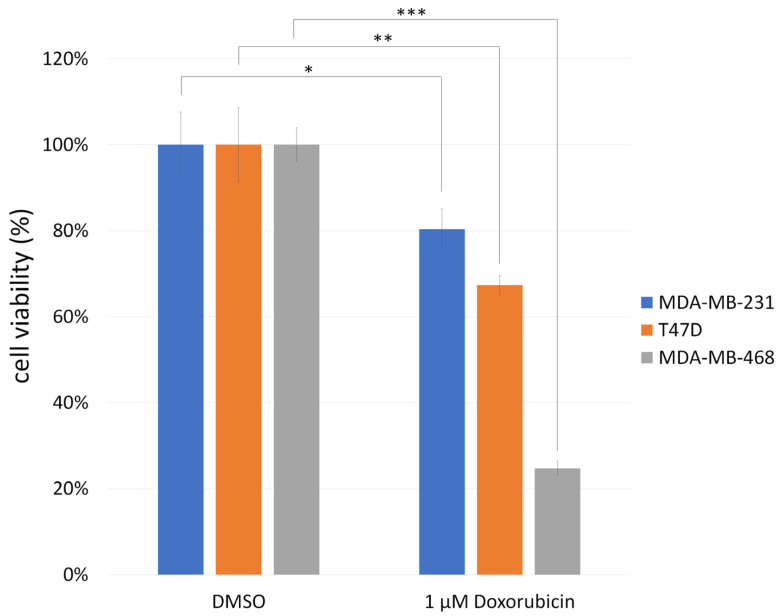
The differences in survival rates between three different breast cancer cell lines after treatment with doxorubicin. MDA-MB-231, T47D, and MDA-MB-468 breast cancer cell lines were treated with 1 μM doxorubicin for 48 h to assess chemotherapeutic response. Following treatment, cell viability was determined using the CCK-8 assay. Viability in the DMSO-treated control group was set as 100% and used as a reference to normalize the results from doxorubicin-treated group. Data are presented as the mean ± standard deviation (SD) from three independent experiments (n = 3). (*, *p* < 0.05; **, *p* < 0.01; ***, *p* < 0.001).

**Figure 4 biomedicines-13-01034-f004:**
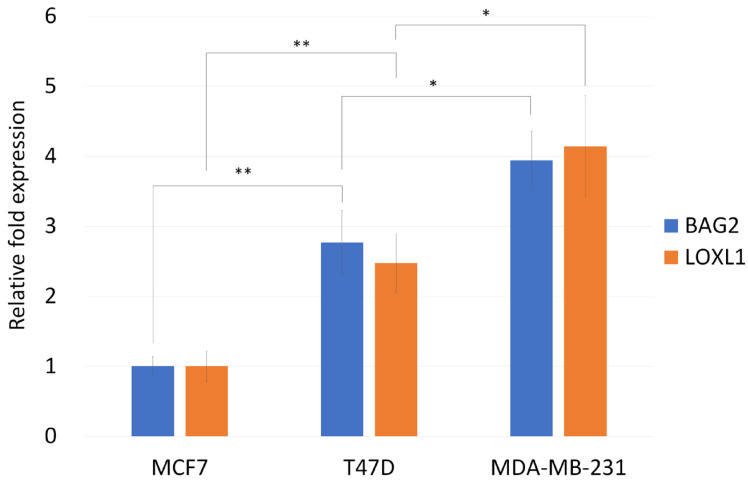
Comparison of the expression levels of BAG2 and LOXL1 in various breast cancer cells. Real-time RT-PCR was used to analyze the expression of BAG2 and LOXL1 in various breast cancer cells. It was found that the expression of both BAG2 and LOXL1 in MDA-MB-231 and T47D was higher than that in MCF7 (the expression of MCF7 was set as 1). The results are displayed as mean ± SD, n = 3 (*, *p* < 0.05; **, *p* < 0.01).

**Figure 5 biomedicines-13-01034-f005:**
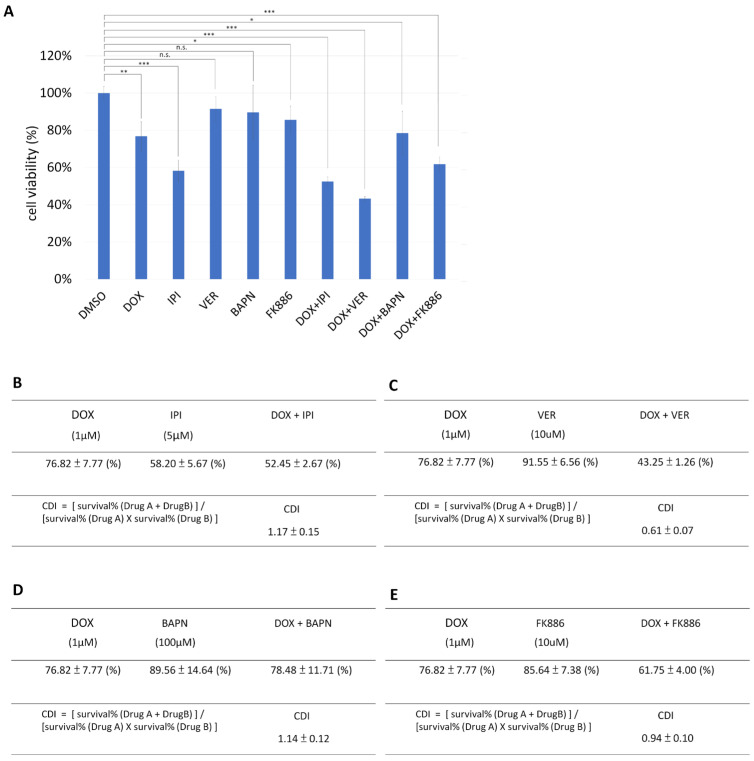
Comparison of the survival rate differences in the breast cancer cell line MDA-MB-231 after co-treatment with various misfolding p53 aggregation inhibitors and doxorubicin. (**A**) Cells were treated with the chemotherapy drug doxorubicin (DOX, 1 μM) and the following concentrations of each misfolding p53 aggregation inhibitor: HSP90 inhibitor (IPI-504, 5 μM), HSP70 inhibitor (VER-155008, 10 μM), LOX inhibitor (BAPN, 100 μM), and NAMPT inhibitor (FK886, 10 μM). After 48 h of drug treatment, cell survival rates were assessed using CCK8 reagent. DMSO only was calculated as 100% to normalize other treatment conditions. Results are presented as mean ± SD, n = 3 (*, *p* < 0.05; **, *p* < 0.01; ***, *p* < 0.001; n.s., *p* > 0.05). (**B**) Calculation of the drug interaction coefficient (CDI) between doxorubicin and HSP90 inhibitor (IPI-504) in MDA-MB-231 cells. According to the cell survival rate calculations from the CCK8 assay, the CDI value is 1.17, indicating that doxorubicin and IPI-504 have no synergistic cytotoxic effect on MDA-MB-231 cells. (**C**) Calculation of CDI for doxorubicin and HSP70 inhibitor (VER-155008) in MDA-MB-231 cells. Based on the cell survival rate calculations from the CCK8 assay, the CDI value is 0.61, indicating that doxorubicin and VER-155008 exhibit a significant synergistic cytotoxic effect on MDA-MB-231 cells. (**D**) Calculation of CDI between doxorubicin and the LOX inhibitor BAPN in MDA-MB-231 cells. According to the CCK8 assay cell survival rate calculations, the CDI value is 1.14, indicating that doxorubicin and BAPN had no drug synergistic cytotoxic effect on MDA-MB-231 cells. (**E**) Calculation of CDI between doxorubicin and FK886 in MDA-MB-231 cells. Based on the cell survival rate calculations from the CCK8 assay, the CDI value is 0.94, indicating that doxorubicin and FK886 had a synergistic cytotoxic effect on MDA-MB-231 cells.

**Figure 6 biomedicines-13-01034-f006:**
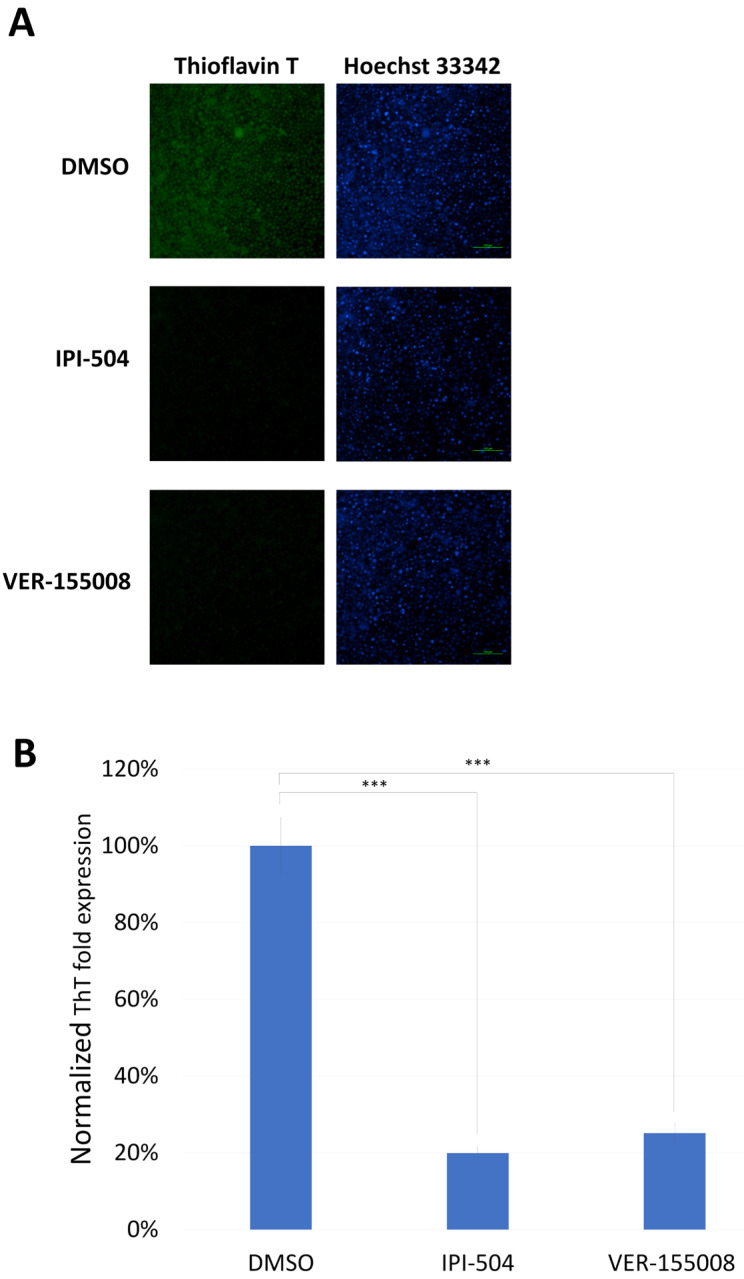
ThT signaling in MDA-MB-231 cancer cells following treatment with IPI-504 or VER-155008. (**A**) MDA-MB-231 cells were treated with DMSO only, 5 μM of the HSP90 inhibitor IPI-504, or 10 μM of the HSP70 inhibitor VER-155008. After 48 h of treatment, the cells were co-stained with ThT and Hoechst 33342. ThT was used to detect protein aggregation (green), while Hoechst 33342 served as a nuclear counterstain (blue). Scale bar: 100 µm (20× magnification). (**B**) The optical density (OD) ratio of ThT to Hoechst 33342 and ThT intensity in the DMSO group were set to 1 for normalization across conditions. As shown, ThT intensity was significantly reduced following treatment with IPI-504 or VER-155008. Data are presented as mean ± SD, n = 3 (***, *p* < 0.001).

**Figure 7 biomedicines-13-01034-f007:**
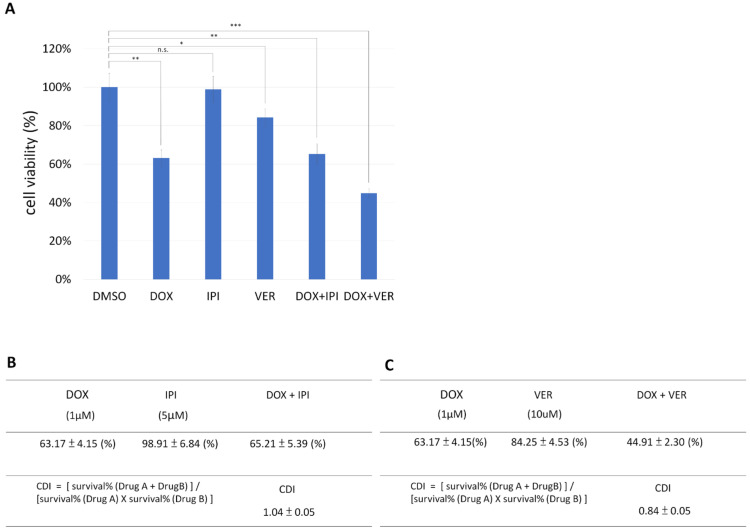
Comparison of the differences in survival rates in the breast cancer cell line T47D after co-treatment with various p53 aggregation inhibitors and doxorubicin. (**A**) Cells were treated with the chemotherapy drug doxorubicin (DOX, 1 μM) and the following concentrations of each p53 aggregation inhibitor drug: HSP90 inhibitor (IPI-504, 5 μM) or HSP70 inhibitor (VER-155008, 10 μM). After drug treatment for 48 h, the cell survival rate was assayed with CCK8 reagent. Cells treated with DMSO alone were used as the baseline control and defined as 100% viability, serving as the reference point for normalizing responses to other drug treatment conditions. Results are displayed as mean ± SD, n = 3 (*, *p* < 0.05; **, *p* < 0.01; ***, *p* < 0.001; n.s., *p* > 0.05). (**B**) Calculation of the drug interaction coefficient (CDI) between doxorubicin and the HSP90 inhibitor IPI-504 in T47D cells. According to the cell survival rate calculation of the CCK8 assay, CDI has a value of 1.04, indicating that doxorubicin and IPI-504 have no drug synergistic cytotoxic effect on T47D cells. (**C**) Calculation of CDI between doxorubicin and HSP70 (VER-155008) in T47D cells. According to the cell survival rate calculation of the CCK8 assay, CDI has a value of 0.84, indicating that doxorubicin and VER-155008 have a drug synergistic cytotoxic effect on T47D cells.

**Figure 8 biomedicines-13-01034-f008:**
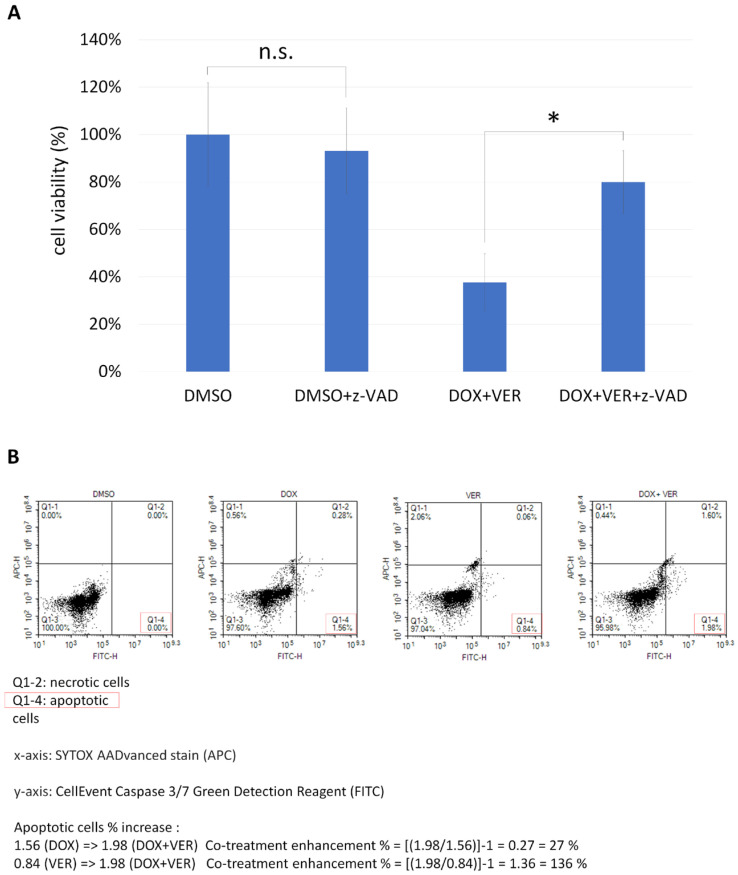
Co-treatment with the HSP70 inhibitor VER-155008 and doxorubicin triggers apoptosis in MDA-MB-231 cells. (**A**) The cytotoxic effect of combining doxorubicin and VER-155008 in MDA-MB-231 cells can be reversed by the pan-caspase inhibitor z-VAD-FMK. The cell survival rate was measured under different conditions: DMSO, z-VAD-FMK (20 µM), DOX (1 µM) + VER (10 µM), and DOX + VER + z-VAD-FMK. DMSO only was used as the baseline (100%) to normalize the results. As shown, z-VAD-FMK alone does not impact cell growth but significantly reduces the cytotoxicity of the DOX + VER combination. Results are displayed as mean ± SD, n = 3 (*, *p* < 0.05; n.s., *p* > 0.05). (**B**) Analysis of apoptosis revealed that the co-treatment of VER and DOX increased the apoptotic cell ratio by 27% compared to doxorubicin treatment alone and by 136% compared to VER treatment alone.

**Figure 9 biomedicines-13-01034-f009:**
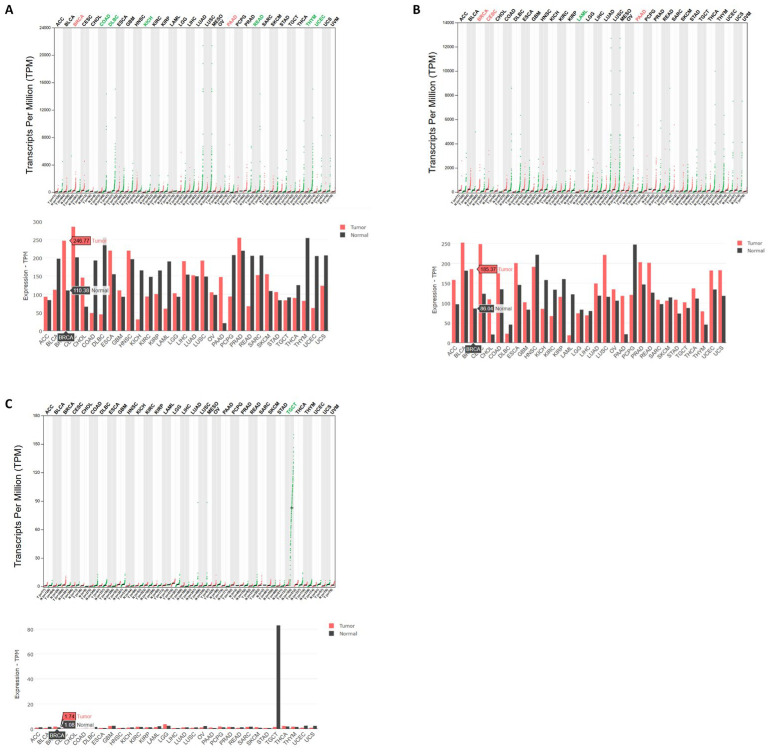
Expression profiles of Hsp70 family members in breast invasive carcinoma (BRCA). (**A**) According to the GEPIA2 database (http://gepia2.cancer-pku.cn/, accessed on 3 April 2025), HSPA1A is significantly highly expressed in BRCA compared to healthy tissue. A red word at the top indicates that the gene expression is significantly increased in cancer samples for this cancer type, while a green word signifies a significant decrease in expression. (**B**) According to the GEPIA2 database (http://gepia2.cancer-pku.cn/, accessed on 3 April 2025), HSPA1B is significantly highly expressed in BRCA compared to healthy tissue. (**C**) According to the GEPIA2 database (http://gepia2.cancer-pku.cn/, accessed on 3 April 2025), HSPA1L is expressed at low levels in all types of cancers but expressed at high levels in normal testis.

## Data Availability

The data supporting the findings of this study are available from the corresponding author upon reasonable request. The data are not publicly available due to containing information that could compromise the privacy concerns.

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
