# Peer review of "Synergistic Anticancer Activity of HSP70 Inhibitor and Doxorubicin in Gain-of-Function Mutated p53 Breast Cancer Cells"

_biomedicines, 2025, doi:10.3390/biomedicines13051034_

Round 1

Reviewer 1 Report

Comments and Suggestions for Authors This manuscript presents a compelling study investigating the synergistic effects of HSP70 inhibition and doxorubicin in breast cancer cells with gain-of-function (GOF) mutated p53. Given the established role of mutant p53 in chemoresistance, the research is timely and relevant. The study employs cell viability assays, drug synergy analysis, apoptosis detection, and aggregation assessment, contributing to our understanding of p53 aggregation-driven resistance and potential therapeutic interventions. While the findings are promising, several key areas require refinement to enhance scientific rigor, clarity, and translational relevance. Specifically, methodological details, statistical reporting, mechanistic insights, and data representation need improvement. Addressing these issues will strengthen the manuscript’s credibility and impact. Major Comments
  1. The authors are suggested to explicitly highlight how this study extends beyond prior research on HSP70 inhibition in p53-mutated cancers. A more precise comparison with existing studies and emphasis on the novel contributions of this work would help establish its unique impact.
  2. The authors are recommended to clarify whether the observed synergistic effect between HSP70 inhibition and doxorubicin is strictly dependent on p53. Since p53-driven chemoresistance is context-dependent, discussing whether alternative pathways might be involved would provide a more comprehensive mechanistic understanding.
  3. The authors are instructed to address how protein aggregation signals were quantified in ThT fluorescence assays. It is unclear how the ratio of ThT to Hoechst 33342 optical density was calculated to normalize the ThT fluorescence intensity. Additionally, the authors should specify the software used for analysis and include this information in both the Methods section and the figure legends.
  4. The authors are recommended to include a detailed section on statistical analysis. Currently, no statistical methods are mentioned in the manuscript, making it unclear how significance was determined. This section should specify the statistical tests used, significance thresholds, and whether multiple comparisons were accounted for.
  5. The authors are advised to provide more precise error bars in all figures. In the current version, error bars are either missing or not well-defined, making it challenging to assess variability. Additionally, statistical significance markers are absent from all figures. Including p-values or significance indicators in figure legends would improve clarity.
  6. The authors are instructed to verify the dot plots presented in Figure 8 carefully. There is a concern that the representative dot plot may not accurately reflect the experimental data, as most cells appear in the viable cell quadrant across all treatment groups, including those expected to induce apoptosis. If an error has occurred, replacing the representative plots with corrected data or providing an explanation in the figure legend is necessary.
  7. The authors are advised to acknowledge that Thioflavin T (ThT) staining alone may not be sufficient to confirm p53 aggregation definitively. If feasible, additional biochemical approaches such as Western blot under denaturing conditions or co-immunoprecipitation (Co-IP) would provide stronger validation. If these experiments are beyond the scope of the study, acknowledging this as a limitation and suggesting future validation strategies would be beneficial.
  8. The authors are recommended to expand on the clinical relevance of the findings. Addressing whether specific patient subgroups, based on HSP70 expression levels or p53 mutation status, would benefit most from this therapy would enhance the study's translational value. If feasible, integrating patient-derived datasets (e.g., TCGA, METABRIC) could provide additional validation for potential biomarker-based patient stratification.
  9. The authors are suggested to discuss pharmacokinetics and toxicity considerations for HSP70 inhibitors. Given that doxorubicin has well-known cardiotoxic effects, discussing whether HSP70 inhibition could allow for dose reduction while maintaining efficacy would be valuable. If pharmacokinetic data for HSP70 inhibitors exist, including these insights would strengthen the translational impact.
  10. The authors are instructed to refine the conclusion to align with the study’s scope. While the data strongly support the synergy between HSP70 inhibition and doxorubicin, broader claims regarding its clinical applicability should be moderated. The authors should focus on the specific conditions under which this approach is most effective and acknowledge the need for further validation in diverse preclinical models.

Author Response

This manuscript presents a compelling study investigating the synergistic effects of HSP70 inhibition and doxorubicin in breast cancer cells with gain-of-function (GOF) mutated p53. Given the established role of mutant p53 in chemoresistance, the research is timely and relevant. The study employs cell viability assays, drug synergy analysis, apoptosis detection, and aggregation assessment, contributing to our understanding of p53 aggregation-driven resistance and potential therapeutic interventions. While the findings are promising, several key areas require refinement to enhance scientific rigor, clarity, and translational relevance. Specifically, methodological details, statistical reporting, mechanistic insights, and data representation need improvement. Addressing these issues will strengthen the manuscript’s credibility and impact. Major Comments

1. The authors are suggested to explicitly highlight how this study extends beyond prior research on HSP70 inhibition in p53-mutated cancers. A more precise comparison with existing studies and emphasis on the novel contributions of this work would help establish its unique impact.

Response: We have added emphasis about how our current study differs from other studies in lines 310-312.

2. The authors are recommended to clarify whether the observed synergistic effect between HSP70 inhibition and doxorubicin is strictly dependent on p53. Since p53-driven chemoresistance is context-dependent, discussing whether alternative pathways might be involved would provide a more comprehensive mechanistic understanding.

Response: The other two papers investigate the HSP70 relief of doxorubicin resistance through Mint3-HIF-1-HSP70 (Cell Death Dis. 2023 Dec 11;14(12):815) and DNAJC12-HSP70-AKT (Redox Biol. 2024 Apr; 70:103035). The mechanism of action of the drug combination involving HSP70 inhibition and doxorubicin is complex and requires further elucidation of other pathways related to p53. Maybe further studies can be conducted to elucidate the upstream and downstream molecule interaction. We have added this point on lines 368-377.

3. The authors are instructed to address how protein aggregation signals were quantified in ThT fluorescence assays. It is unclear how the ratio of ThT to Hoechst 33342 optical density was calculated to normalize the ThT fluorescence intensity. Additionally, the authors should specify the software used for analysis and include this information in both the Methods section and the figure legends.

Response: We have added more detailed information about quantification in the ThT fluorescence assays in the Methods section.

4. The authors are recommended to include a detailed section on statistical analysis. Currently, no statistical methods are mentioned in the manuscript, making it unclear how significance was determined. This section should specify the statistical tests used, significance thresholds, and whether multiple comparisons were accounted for.

Response: Thank you for the comment. We have added the statistical methods used in the Methods section and Figures.

5. The authors are advised to provide more precise error bars in all figures. In the current version, error bars are either missing or not well-defined, making it challenging to assess variability. Additionally, statistical significance markers are absent from all figures. Including p-values or significance indicators in figure legends would improve clarity.

Response: We have added the statistics with p-values or significance indicators in the Figures.

6. The authors are instructed to verify the dot plots presented in Figure 8 carefully. There is a concern that the representative dot plot may not accurately reflect the experimental data, as most cells appear in the viable cell quadrant across all treatment groups, including those expected to induce apoptosis. If an error has occurred, replacing the representative plots with corrected data or providing an explanation in the figure legend is necessary.

Response: We previously used the same kit to stain the apoptotic caspase-3/7(+) SYTOX(-), and we did not obtain a strong signal then either (Figure 8B in Biochim Biophys Acta Mol Basis Dis. 2025 Mar;1871(3):167662). This may be because CellEvent Caspase-3/7 Green is much less sensitive than CellEvent Caspase-3/7 Red. We used CellEvent Caspase-3/7 Red as in Figure 8A in the same BBA paper. The new Figure S1 shows the result using CellEvent Caspase-3/7 Red, which may strengthen the validity of the results in Fig. 8B.

7. The authors are advised to acknowledge that Thioflavin T (ThT) staining alone may not be sufficient to confirm p53 aggregation definitively. If feasible, additional biochemical approaches such as Western blot under denaturing conditions or co-immunoprecipitation (Co-IP) would provide stronger validation. If these experiments are beyond the scope of the study, acknowledging this as a limitation and suggesting future validation strategies would be beneficial.

Response: We previously performed p53 and Thioflavin T colonization to assess the degree of p53 aggregation in head and neck cancer cells (Figure 4B in Biomolecules. 2022 Mar 12;12(3):438. & Figure S1 in Biochim Biophys Acta Mol Basis Dis. 2025 Mar;1871(3):167662.). We have added the Thioflavin T and DO-1 p53 monoclonal antibody co-staining in breast cancer cells, as shown in Figure S1.

8. The authors are recommended to expand on the clinical relevance of the findings. Addressing whether specific patient subgroups, based on HSP70 expression levels or p53 mutation status, would benefit most from this therapy would enhance the study's translational value. If feasible, integrating patient-derived datasets (e.g., TCGA, METABRIC) could provide additional validation for potential biomarker-based patient stratification.

Response: We have added the information in lines 312-329.

9. The authors are suggested to discuss pharmacokinetics and toxicity considerations for HSP70 inhibitors. Given that doxorubicin has well-known cardiotoxic effects, discussing whether HSP70 inhibition could allow for dose reduction while maintaining efficacy would be valuable. If pharmacokinetic data for HSP70 inhibitors exist, including these insights would strengthen the translational impact.

Response: We added some information in lines 378-384.

10. The authors are instructed to refine the conclusion to align with the study’s scope. While the data strongly support the synergy between HSP70 inhibition and doxorubicin, broader claims regarding its clinical applicability should be moderated. The authors should focus on the specific conditions under which this approach is most effective and acknowledge the need for further validation in diverse preclinical models.

Response: We have modified the discussion and conclusion in lines 381-391.

Reviewer 2 Report

Comments and Suggestions for Authors

This study investigates how specific p53 mutations in breast cancer form prion-like aggregates, leading to doxorubicin resistance and suggest that targeting HSP70 can disrupt mutant p53 aggregation, restoring doxorubicin sensitivity in resistant breast cancers. However, the dual-drug combination, which is not novel, is the main reason why this article is not innovative enough. In addition, the proofreading and statistics of the text and corresponding figures need to be further improved, which is the biggest limitation of this article.

Major points:

  1. Since many papers have reported the combined effects of HSP70 inhibitors and doxorubicin, the main shortcoming of this paper is that the authors did not fully demonstrate the innovation and advantages compared with other similar studies.
  2. How did the authors determine the concentration when using various p53 aggregation inhibitors and doxorubicin to treat different breast cancer cell lines? Each cell line has different sensitivity to drugs. The authors should at least explore the concentration gradient of a single drug to better illustrate the significance of the dual drug combination.
  3. As a caspase activator, z-VAD-FMK should be effectively demonstrated by WB in a variety of ways to illustrate the inhibitory effect of z-VAD-FMK and the caspase cleavage induced by the combination of two drugs.
  4. The manuscript and figures need more rigorous editing. For example, Fig2A and Fig6A lack bars, and the bar graphs of each figure lack statistics; and the representative flow cytometry graph of Fig8B does not seem to show significant differences. It is very necessary to add statistical data here.
  5. This paper is written in a very simplified way, and the research background and scientific logic are not well organized. Therefore, readers may find it difficult to understand the principles behind the experiments conducted. Authors are strongly encouraged to thoroughly revise their manuscripts, reorganize the content for clarity, and seek the help of a professional scientific English editor to ensure coherence and accuracy.

Comments on the Quality of English Language

Need to be improved

Author Response

This study investigates how specific p53 mutations in breast cancer form prion-like aggregates, leading to doxorubicin resistance and suggest that targeting HSP70 can disrupt mutant p53 aggregation, restoring doxorubicin sensitivity in resistant breast cancers. However, the dual-drug combination, which is not novel, is the main reason why this article is not innovative enough. In addition, the proofreading and statistics of the text and corresponding figures need to be further improved, which is the biggest limitation of this article.

Major points:

1. Since many papers have reported the combined effects of HSP70 inhibitors and doxorubicin, the main shortcoming of this paper is that the authors did not fully demonstrate the innovation and advantages compared with other similar studies.

Response: The objective of this study was to determine which p53 aggregation inhibitor had the most significant effect, aiming to resolve doxorubicin drug resistance in gain-of-function (GOF) p53-mutated breast cancer cells. Recently, a report mentioned that the use of a Hsp90 inhibitor could resolve the misfolded p53 aggregation in p53-mutated breast cancer cells, thereby overcoming doxorubicin drug resistance (Theranostics 2023, 13, 339-354). However, our screening of aggregation inhibitors revealed that HSP70 inhibitors appear to be more effective than others, including Hsp90 inhibitors. Using the keywords “HSP70 inhibitor doxorubicin breast cancer” led to the discovery of two papers that suggested that inhibiting HSP70 also reversed doxorubicin resistance in breast cancer. (1. Cell Death Dis. 2023 Dec 11;14(12):815. 2. Redox Biol. 2024 Apr:70:103035.) However, these two papers did not investigate the aggregation of p53 GOF in response to the same HSP70 inhibitor and doxorubicin combination treatment.

2. How did the authors determine the concentration when using various p53 aggregation inhibitors and doxorubicin to treat different breast cancer cell lines? Each cell line has different sensitivity to drugs. The authors should at least explore the concentration gradient of a single drug to better illustrate the significance of the dual drug combination.

Response: We used several mutated p53 aggregation inhibitors in head and neck cancer cells, as previously reported in our paper in Biomolecules. 2022 Mar 12;12(3):438. & Biochim Biophys Acta Mol Basis Dis. 2025 Mar;1871(3):167662. Because this paper used a lot of dual drug combination pairing experiments, performing all the single-drug gradients would be a big task. Therefore, we just followed the same concentration as in the head and neck cancer cells.

3. As a caspase activator, z-VAD-FMK should be effectively demonstrated by WB in a variety of ways to illustrate the inhibitory effect of z-VAD-FMK and the caspase cleavage induced by the combination of two drugs.

Response: Our lab had no caspase antibody, so we ordered the caspase 3 antibody on the day after receiving the first-round decision about the manuscript. But it took 2~3 weeks to arrive at the lab, and it was past the deadline for resubmission. We previously used the CellEvent Caspase-3/7 Red Detection Reagent (Invitrogen) to study Caspase-3/7 activation in head and neck cancer cells. Therefore, we conducted a new experiment to assay Caspase-3/7 activation under drug treatment in breast cancer cells and have added the results to the manuscript as Figure S1.

4. The manuscript and figures need more rigorous editing. For example, Fig2A and Fig6A lack bars, and the bar graphs of each figure lack statistics; and the representative flow cytometry graph of Fig8B does not seem to show significant differences. It is very necessary to add statistical data here.

Response: We thank the reviewer for the comment. We have added the statistics to the Figures. We previously used the same kit to stain the apoptotic caspase-3/7(+) SYTOX(-). We did not get a strong signal at that time either (Figure 8B in Biochim Biophys Acta Mol Basis Dis. 2025 Mar;1871(3):167662). This may be because CellEvent Caspase-3/7 Green is much less sensitive than CellEvent Caspase-3/7 Red. We used CellEvent Caspase-3/7 Red as in Figure 8A in the same BBA paper. The results in the new Figure S1 used CellEvent Caspase-3/7 Red, and the new result may validate the results of Fig. 8B.

5. This paper is written in a very simplified way, and the research background and scientific logic are not well organized. Therefore, readers may find it difficult to understand the principles behind the experiments conducted. Authors are strongly encouraged to thoroughly revise their manuscripts, reorganize the content for clarity, and seek the help of a professional scientific English editor to ensure coherence and accuracy.

Response:  We have made various revisions and added new information to fulfill the reviewer's requests, and furthermore performed a second round of English editing. This article has been fully edited by a professional, certified scientific English editor.

Round 2

Reviewer 1 Report

Comments and Suggestions for Authors

The authors efficiently addressed all the concerns in the revised manuscript, which is suitable for publication in Biomedicines in its current form.

Reviewer 2 Report

Comments and Suggestions for Authors

The author has solved the problem I raised